# Eluder Dimension and the Sample Complexity of Optimistic Exploration

**Daniel Russo**
Stanford University
Stanford, CA 94305
djrusso@stanford.edu

**Benjamin Van Roy**
Stanford University
Stanford, CA 94305
bvr@stanford.edu

## Abstract

This paper considers the sample complexity of the multi-armed bandit with dependencies among the arms. Some of the most successful algorithms for this problem use the principle of optimism in the face of uncertainty to guide exploration. The clearest example of this is the class of *upper confidence bound* (UCB) algorithms, but recent work has shown that a simple posterior sampling algorithm, sometimes called *Thompson sampling*, can be analyzed in the same manner as optimistic approaches. In this paper, we develop a regret bound that holds for both classes of algorithms. This bound applies broadly and can be specialized to many model classes. It depends on a new notion we refer to as the *eluder dimension*, which measures the degree of dependence among action rewards. Compared to UCB algorithm regret bounds for specific model classes, our general bound matches the best available for linear models and is stronger than the best available for generalized linear models.

## 1   Introduction

Consider a politician trying to elude a group of reporters. She hopes to keep her true position hidden from the reporters, but each piece of information she provides must be new, in the sense that it's not a clear consequence of what she has already told them. How long can she continue before her true position is pinned down? This is the essence of what we call the *eluder dimension*. We show this notion controls the rate at which algorithms using *optimistic* exploration converge to optimality.

We consider an optimization problem faced by an agent who is uncertain about how her actions influence performance. The agent selects actions sequentially, and upon each action observes a reward. A *reward function* governs the mean reward of each action. As rewards are observed the agent learns about the reward function, and this allows her to improve behavior. Good performance requires adaptively sampling actions in a way that strikes an effective balance between exploring poorly understood actions and exploiting previously acquired knowledge to attain high rewards.

Unless the agent has prior knowledge of the structure of the mean payoff function, she can only learn to attain near optimal performance by exhaustively sampling each possible action. In this paper, we focus on problems where there is a known relationship among the rewards generated by different actions, potentially allowing the agent to learn without exploring every action. Problems of this form are often referred to as multi-armed bandit (MAB) problems with dependent arms.

A notable example is the "linear bandit" problem, where actions are described by a finite number of features and the reward function is linear in these features. Several researchers have studied algorithms for such problems and established theoretical guarantees that have *no dependence* on the number of actions [1, 2, 3]. Instead, their bounds depend on the linear dimension of the class of reward functions. In this paper, we assume that the reward function lies in a known but otherwise arbitrary class of uniformly bounded real-valued functions, and provide theoretical guarantees that

depend on more general measures of the complexity of the class of functions. Our analysis of this abstract framework yields a result that applies broadly, beyond the scope of specific problems that have been studied in the literature, and also identifies fundamental insights that unify more specialized prior results.

The guarantees we provide apply to two popular classes of algorithms for the stochastic MAB: *upper confidence bound* (UCB) algorithms and *Thompson sampling*. Each algorithm is described in Section 3. The aforementioned papers on the linear bandit problem study UCB algorithms [1, 2, 3]. Other authors have studied UCB algorithms in cases where the reward function is Lipschitz continuous [4, 5], sampled from a Gaussian process [6], or takes the form of a generalized [7] or sparse [8] linear model. More generally, there is an immense literature on this approach to balancing between exploration and exploitation, including work on bandits with independent arms [9, 10, 11, 12], reinforcement learning [13, 14], and Monte Carlo Tree Search [15].

Recently, a simple posterior sampling algorithm called *Thompson sampling* was shown to share a close connection with UCB algorithms [16]. This connection enables us to study both types of algorithms in a unified manner. Though it was first proposed in 1933 [17], Thompson sampling has until recently received relatively little attention. Interest in the algorithm grew after empirical studies [18, 19] demonstrated performance exceeding state-of the-art methods. Strong theoretical guarantees are now available for an important class of problems with independent arms [20, 21, 22]. A recent paper considers the application of this algorithm to a linear contextual bandit problem [23].

To our knowledge, few other papers have studied MAB problems in a general framework like the one we consider. There is work that provides general bounds for contextual bandit problems where the context space is allowed to be infinite, but the action space is small (see e.g., [24]). Our model captures contextual bandits as a special case, but we emphasize problem instances with large or infinite action sets, and where the goal is to learn without sampling every possible action. The closest related work to ours is that of Amin et al. [25], who consider the problem of learning the optimum of a function that lies in a known, but otherwise arbitrary set of functions. They provide bounds based on a new notion of dimension, but unfortunately this notion does not provide a guarantee for the algorithms we consider.

We provide bounds on expected regret over a time horizon $T$ that are, up to a logarithmic factor, of order

$$\sqrt{\underbrace{\dim_E\left(\mathcal{F}, T^{-2}\right)}_{\text{Eluder dimension}} \underbrace{\log\left(N\left(\mathcal{F}, T^{-2}, \|\cdot\|_\infty\right)\right)}_{\text{log–covering number}} T}.$$

This quantity depends on the class of reward functions $\mathcal{F}$ through two measures of complexity. Each captures the approximate structure of the class of functions at a scale $T^{-2}$ that depends on the time horizon. The first measures the growth rate of the covering numbers of $\mathcal{F}$, and is closely related to measures of complexity that are common in the supervised learning literature. This quantity roughly captures the sensitivity of $\mathcal{F}$ to statistical over-fitting. The second measure, the *eluder dimension*, is a new notion we introduce. This captures how effectively the value of unobserved actions can be inferred from observed samples. We highlight in Section 4.1 why notions of dimension common to the supervised learning literature are insufficient for our purposes. Finally, we show that our more general result when specialized to linear models recovers the strongest known regret bound and in the case of generalized linear models yields a bound stronger than that established in prior literature.

## 2 Problem Formulation

We consider a model involving a set of actions $\mathcal{A}$ and a set of real-valued functions $\mathcal{F} = \{f_\rho : \mathcal{A} \mapsto \mathbb{R} \,|\, \rho \in \Theta\}$, indexed by a parameter that takes values from an index set $\Theta$. We will define random variables with respect to a probability space $(\Omega, \mathbb{F}, \mathbb{P})$. A random variable $\theta$ indexes the true reward function $f_\theta$. At each time $t$, the agent is presented with a possibly random subset $\mathcal{A}_t \subseteq \mathcal{A}$ and selects an action $A_t \in \mathcal{A}_t$, after which she observes a reward $R_t$.

We denote by $H_t$ the history $(\mathcal{A}_1, A_1, R_1, \ldots, \mathcal{A}_{t-1}, A_{t-1}, R_{t-1}, \mathcal{A}_t)$ of observations available to the agent when choosing an action $A_t$. The agent employs a policy $\pi = \{\pi_t | t \in \mathbb{N}\}$, which is a deterministic sequence of functions, each mapping the history $H_t$ to a probability distribution over actions $\mathcal{A}$. For each realization of $H_t$, $\pi_t(H_t)$ is a distribution over $\mathcal{A}$ with support $\mathcal{A}_t$. The action $A_t$

is selected by sampling from the distribution $\pi_t(\cdot)$, so that $\mathbb{P}(A_t \in \cdot | H_t) = \pi_t(H_t)$. We assume that $\mathbb{E}[R_t | H_t, \theta, A_t] = f_\theta(A_t)$. In other words, the realized reward is the mean-reward value corrupted by zero-mean noise. We will also assume that for each $f \in \mathcal{F}$ and $t \in \mathbb{N}$, $\arg\max_{a \in \mathcal{A}_t} f(a)$ is nonempty with probability one, though algorithms and results can be generalized to handle cases where this assumption does not hold. We fix constants $C > 0$ and $\eta > 0$ and impose two further simplifying assumptions. The first concerns boundedness of reward functions.

**Assumption 1.** *For all $f \in \mathcal{F}$ and $a \in \mathcal{A}$, $f(a) \in [0, C]$.*

Our second assumption ensures that observation noise is light-tailed. We say a random variable $X$ is $\eta$-sub-Gaussian if $\mathbb{E}[\exp(\lambda X)] \leq \exp(\lambda^2 \eta^2 / 2)$ almost surely for all $\lambda$.

**Assumption 2.** *For all $t \in \mathbb{N}$, $R_t - f_\theta(A_t)$ conditioned on $(H_t, \theta, A_t)$ is $\eta$-sub-Gaussian.*

We let $A_t^* \in \arg\max_{a \in \mathcal{A}_t} f_\theta(a)$ denote the optimal action at time $t$. The $T$ period *regret* is the random variable

$$\mathcal{R}(T, \pi) = \sum_{t=1}^{T} [f_\theta(A_t^*) - f_\theta(A_t)],$$

where the actions $\{A_t : t \in \mathbb{N}\}$ are selected according to $\pi$. We sometimes study expected regret $\mathbb{E}[\mathcal{R}(T, \pi)]$, where the expectation is taken over the prior distribution of $\theta$, the reward noise, and the algorithm's internal randomization. This quantity is sometimes called *Bayes risk* or *Bayesian regret*. Similarly, we study conditional expected regret $\mathbb{E}[\mathcal{R}(T, \pi) | \theta]$, which integrates over all randomness in the system except for $\theta$.

**Example 1. Contextual Models.** *The contextual multi-armed bandit model is a special case of the formulation presented above. In such a model, an exogenous Markov process $X_t$ taking values in a set $\mathcal{X}$ influences rewards. In particular, the expected reward at time $t$ is given by $f_\theta(a, X_t)$. However, this is mathematically equivalent to a problem with stochastic time-varying decision sets $\mathcal{A}_t$. In particular, one can define the set of actions to be the set of state-action pairs $\mathcal{A} := \{(x, a) : x \in \mathcal{A}, a \in \mathcal{A}(x)\}$, and the set of available actions to be $\mathcal{A}_t = \{(X_t, a) : a \in \mathcal{A}(X_t)\}$.*

## 3 Algorithms

We will establish performance bounds for two classes of algorithms: Thompson sampling and UCB algorithms. As background, we discuss the algorithms in this section. We also provide an example of each type of algorithm that is designed to address the "linear bandit" problem.

**UCB Algorithms:** UCB algorithms have received a great deal of attention in the MAB literature. Here we describe a very broad class of UCB algorithms. We say that a *confidence set* is a random subset $\mathcal{F}_t \subset \mathcal{F}$ that is measurable with respect to $\sigma(H_t)$. Typically, $\mathcal{F}_t$ is constructed so that it contains $f_\theta$ with high probability. We denote by $\pi^{\mathcal{F}_{1:\infty}}$ a UCB algorithm that makes use of a sequence of confidence sets $\{\mathcal{F}_t : t \in \mathbb{N}\}$. At each time $t$, such an algorithm selects the action

$$A_t \in \arg\max_{a \in \mathcal{A}_t} \sup_{f \in \mathcal{F}_t} f(a),$$

where $\sup_{f \in \mathcal{F}_t} f(a)$ is an optimistic estimate of $f_\theta(a)$ representing the greatest value that is statistically plausible at time $t$. Optimism encourages selection of poorly-understood actions, which leads to informative observations. As data accumulates, optimistic estimates are adapted, and this process of exploration and learning converges toward optimal behavior.

In this paper, we will assume for simplicity that the maximum defining $A_t$ is attained. Results can be generalized to handle cases when this technical condition does not hold. Unfortunately, for natural choices of $\mathcal{F}_t$, it may be exceptionally difficult to solve for such an action. Thankfully, all results in this paper also apply to a posterior sampling algorithm that avoids this hard optimization problem.

**Thompson sampling:** The Thompson sampling algorithm simply samples each action according to the probability it is optimal. In particular, the algorithm applies action sampling distributions $\pi_t^{\text{TS}}(H_t) = \mathbb{P}(A_t^* \in \cdot | H_t)$, where $A_t^*$ is a random variable that satisfies $A_t^* \in \arg\max_{a \in \mathcal{A}_t} f_\theta(a)$. Practical implementations typically operate by at each time $t$ sampling an index $\hat{\theta}_t \in \Theta$ from the distribution $\mathbb{P}(\theta \in \cdot | H_t)$ and then generating an action $A_t \in \arg\max_{a \in \mathcal{A}_t} f_{\hat{\theta}_t}(a)$.

**Algorithm 1** Linear UCB

1: **Initialize**: Select $d$ linearly independent actions
2: **Update Statistics**:
   $\hat{\theta}_t \leftarrow$ OLS estimate of $\theta$
   $\Phi_t \leftarrow \sum_{k=1}^{t-1} \phi(\bar{A}_k)\phi(\bar{A}_k)^T$
   $\Theta_t \leftarrow \left\{ \rho : \left\| \rho - \hat{\theta}_t \right\|_{\Phi_t} \leq \beta\sqrt{d\log t} \right\}$
3: **Select Action**:
   $\overline{A}_t \in \arg\max_{a \in \mathcal{A}} \left\{ \max_{\rho \in \Theta_t} \langle \phi(a), \rho \rangle \right\}$
4: **Increment $t$ and Goto Step 2**

**Algorithm 2**
Linear Thompson sampling

1: **Sample Model**:
   $\hat{\theta}_t \sim N(\mu_t, \Sigma_t)$
2: **Select Action**:
   $A_t \in \arg\max_{a \in \mathcal{A}} \langle \phi(a), \hat{\theta}_t \rangle$
3: **Update Statistics**:
   $\mu_{t+1} \leftarrow \mathbb{E}[\theta|H_{t+1}]$
   $\Sigma_{t+1} \leftarrow \mathbb{E}[(\theta - \mu_{t+1})(\theta - \mu_{t+1})^\top|H_{t+1}]$
4: **Increment $t$ and Goto Step 1**

**Algorithms for Linear Bandits:** Here we provide an example of a Thompson sampling and a UCB algorithm, each of which addresses a problem in which the reward function is linear in a $d$-dimensional vector $\theta$. In particular, there is a known feature mapping $\phi : \mathcal{A} \to \mathbb{R}^d$ such that an action $a$ yields expected reward $f_\theta(a) = \langle \phi(a), \theta \rangle$. Algorithm 1 is a variation of one proposed by Rusmevichientong and Tsitsiklis [3] to address such problems. Given past observations, the algorithm constructs a confidence ellipsoid $\Theta_t$ centered around a least squares estimate $\hat{\theta}_t$ and employs the upper confidence bound $U_t(a) := \max_{\overline{\theta} \in \Theta_t} \langle \phi(a), \overline{\theta} \rangle = \langle \phi(a), \hat{\theta}_t \rangle + \beta\sqrt{d\log(t)} \|\phi(a)\|_{\Phi_t^{-1}}$. The term $\|\phi(a)\|_{\Phi_t^{-1}}$ captures the amount of previous exploration in the direction $\phi(a)$, and causes the "uncertainty bonus" $\beta\sqrt{d\log(t)} \|\phi(a)\|_{\Phi_t^{-1}}$ to diminish as the number of observations increases.

Now, consider Algorithm 2. Here we assume $\theta$ is drawn from a normal distribution $N(\mu_1, \Sigma_1)$. We consider a linear reward function $f_\theta(a) = \langle \phi(a), \theta \rangle$ and assume the reward noise $R_t - f_\theta(A_t)$ is normally distributed and independent from $(H_t, A_t, \theta)$. It is easy to show that, conditioned on the history $H_t$, $\theta$ remains normally distributed. Algorithm 2 presents an implementation of Thompson sampling for this problem. The expectations can be computed efficiently via Kalman filtering.

# 4 Notions of Dimension

Recently, there has been a great deal of interest in the development of regret bounds for linear UCB algorithms [1, 2, 3, 26]. These papers show that for a broad class of problems, a variant $\pi^*$ of Algorithm 1 satisfies the upper bounds $\mathbb{E}[\mathcal{R}(T, \pi^*)] = \tilde{O}(d\sqrt{T})$ and $\mathbb{E}[\mathcal{R}(T, \pi^*) \mid \theta] = \tilde{O}(d\sqrt{T})$. An interesting feature of these bounds is that they have no dependence on the number actions in $\mathcal{A}$, and instead depend only on the *linear dimension* of the set of functions $\mathcal{F}$. Our goal is to provide bounds that depend on more general measures of the complexity of the class of functions. This section introduces a new notion, the *eluder dimension*, on which our bounds will depend. First, we highlight why common notions from statistical learning theory do not suffice when it comes to multi–armed bandit problems.

## 4.1 Vapnik-Chervonenkis Dimension

We begin with an example that illustrates how a class of functions that is learnable in constant time in a supervised learning context may require an arbitrarily long duration when learning to optimize.

**Example 2.** *Consider a finite class of binary-valued functions $\mathcal{F} = \{f_\rho : \mathcal{A} \mapsto \{0, 1\} \mid \rho \in \{1, \ldots, n\}\}$ over a finite action set $\mathcal{A} = \{1, \ldots, n\}$. Let $f_\rho(a) = \mathbf{1}(\rho = a)$, so that each function is an indicator for an action. To keep things simple, assume that $R_t = f_\theta(A_t)$, so that there is no noise. If $\theta$ is uniformly distributed over $\{1, \ldots, n\}$, it is easy to see that the regret of any algorithm grows linearly with $n$. For large $n$, until $\theta$ is discovered, each sampled action is unlikely to reveal much about $\theta$ and learning therefore takes very long.*

*Consider the closely related supervised learning problem in which at each time an action $\tilde{A}_t$ is sampled uniformly from $\mathcal{A}$ and the mean–reward value $f_\theta(\tilde{A}_t)$ is observed. For large $n$, the time it*

*takes to effectively learn to predict $f_\theta(\tilde{A}_t)$ given $\tilde{A}_t$ does not depend on $t$. In particular, prediction error converges to $1/n$ in constant time. Note that predicting $0$ at every time already achieves this low level of error.*

In the preceding example, the Vapnik-Chervonenkis (VC) dimension, which characterizes the sample complexity of supervised learning, is $1$. On the other hand, the eluder-dimension, which will we define below, is $n$. To highlight conceptual differences between the eluder dimension and the VC dimension, we will now define VC dimension in a way analogous to how will define eluder dimension. We begin with a notion of independence.

**Definition 1.** An action $a$ is *VC-independent* of $\tilde{\mathcal{A}} \subseteq \mathcal{A}$ if for any $f$, $\tilde{f} \in \mathcal{F}$ there exists some $\bar{f} \in \mathcal{F}$ which agrees with $f$ on $a$ and with $\tilde{f}$ on $\tilde{\mathcal{A}}$; that is, $\bar{f}(a) = f(a)$ and $\bar{f}(\tilde{a}) = \tilde{f}(\tilde{a})$ for all $\tilde{a} \in \tilde{\mathcal{A}}$. Otherwise, $a$ is *VC-dependent* on $\tilde{\mathcal{A}}$.

By this definition, an action $a$ is said to be VC-dependent on $\tilde{\mathcal{A}}$ if knowing the values $f \in \mathcal{F}$ takes on $\tilde{\mathcal{A}}$ *could restrict* the set of possible values at $a$. This notion of independence is intimately related to the VC dimension of a class of functions. In fact, it can be used to define VC dimension.

**Definition 2.** The VC dimension of a class of binary-valued functions with domain $\mathcal{A}$ is the largest cardinality of a set $\tilde{\mathcal{A}} \subseteq \mathcal{A}$ such that every $a \in \tilde{\mathcal{A}}$ is VC-independent of $\tilde{\mathcal{A}} \backslash \{a\}$.

In the above example, any two actions are VC-dependent because knowing the label $f_\theta(a)$ of one action could completely determine the value of the other action. However, this only happens if the sampled action has label $1$. If it has label $0$, one cannot infer anything about the value of the other action. Instead of capturing the fact that one *could* gain useful information through exploration, we need a stronger requirement that guarantees one *will* gain useful information.

## 4.2 Defining Eluder Dimension

Here we define the eluder dimension of a class of functions, which plays a key role in our results.

**Definition 3.** An action $a \in \mathcal{A}$ is $\epsilon$-*dependent* on actions $\{a_1, ..., a_n\} \subseteq \mathcal{A}$ with respect to $\mathcal{F}$ if any pair of functions $f, \tilde{f} \in \mathcal{F}$ satisfying $\sqrt{\sum_{i=1}^n (f(a_i) - \tilde{f}(a_i))^2} \leq \epsilon$ also satisfies $f(a) - \tilde{f}(a) \leq \epsilon$. Further, $a$ is $\epsilon$-independent of $\{a_1, .., a_n\}$ with respect to $\mathcal{F}$ if $a$ is not $\epsilon$-dependent on $\{a_1, .., a_n\}$.

Intuitively, an action $a$ is independent of $\{a_1, ..., a_n\}$ if two functions that make similar predictions at $\{a_1, ..., a_n\}$ can nevertheless differ significantly in their predictions at $a$. The above definition measures the "similarity" of predictions at $\epsilon$-scale, and measures whether two functions make similar predictions at $\{a_1, ..., a_n\}$ based on the cumulative discrepancy $\sqrt{\sum_{i=1}^n (f(a_i) - \tilde{f}(a_i))^2}$. This measure of dependence suggests using the following notion of dimension.

**Definition 4.** The $\epsilon$-*eluder dimension* $\dim_E(\mathcal{F}, \epsilon)$ is the length $d$ of the longest sequence of elements in $\mathcal{A}$ such that, for some $\epsilon' \geq \epsilon$, every element is $\epsilon'$-independent of its predecessors.

Recall that a vector space has dimension $d$ if and only if $d$ is the length of the longest sequence of elements such that each element is linearly independent or equivalently, $0$-independent of its predecessors. Definition 4 replaces the requirement of linear independence with $\epsilon$-independence. This extension is advantageous as it captures both nonlinear dependence and approximate dependence.

## 5 Confidence Bounds and Regret Decompositions

A key to our analysis is recent observation [16] that the regret of both Thompson sampling and a UCB algorithm can be decomposed in terms of confidence sets. Define the width of a subset $\tilde{\mathcal{F}} \subset \mathcal{F}$ at an action $a \in \mathcal{A}$ by

$$w_{\tilde{\mathcal{F}}}(a) = \sup_{\underline{f}, \overline{f} \in \tilde{\mathcal{F}}} \left( \overline{f}(a) - \underline{f}(a) \right). \tag{1}$$

This is a worst–case measure of the uncertainty about the payoff $f_\theta(a)$ at $a$ given that $f_\theta \in \tilde{\mathcal{F}}$.

**Proposition 1.** *Fix any sequence $\{\mathcal{F}_t : t \in \mathbb{N}\}$, where $\mathcal{F}_t \subset \mathcal{F}$ is measurable with respect to $\sigma(H_t)$. Then for any $T \in \mathbb{N}$, with probability 1,*

$$\mathcal{R}(T, \pi^{\mathcal{F}_{1:\infty}}) \quad \leq \quad \sum_{t=1}^{T} [w_{\mathcal{F}_t}(A_t) + C\mathbf{1}(f_\theta \notin \mathcal{F}_t)] \tag{2}$$

$$\mathbb{E}\left[\mathcal{R}(T, \pi^{\mathrm{TS}})\right] \quad \leq \quad \mathbb{E}\sum_{t=1}^{T} [w_{\mathcal{F}_t}(A_t) + C\mathbf{1}(f_\theta \notin \mathcal{F}_t)]. \tag{3}$$

If the confidence sets $\mathcal{F}_t$ are constructed to contain $f_\theta$ with high probability, this proposition essentially bounds regret in terms of the sum of widths $\sum_{t=1}^{T} w_{\mathcal{F}_t}(A_t)$. In this sense, the decomposition bounds regret only in terms of uncertainty about the actions $A_1,...,A_t$ that the algorithm has actually sampled. As actions are sampled, the value of $f_\theta(\cdot)$ at those actions is learned accurately, and hence we expect that the width $w_{\mathcal{F}_t}(\cdot)$ of the confidence sets should diminish over time.

It is worth noting that the regret bound of the UCB algorithm $\pi^{\mathcal{F}_{1:\infty}}$ depends on the specific confidence sets $\{\mathcal{F}_t : t \in \mathbb{N}\}$ used by the algorithm whereas the bound of $\pi^{\mathrm{TS}}$ applies for *any* sequence of confidence sets. However, the decomposition (3) holds only in expectation under the prior distribution. The implications of these decompositions are discussed further in earlier work [16].

In the next section, we design abstract confidence sets $\mathcal{F}_t$ that are shown to contain the true function $f_\theta$ with high probability. Then, in Section 7 we give a worst case bound on the sum $\sum_{t=1}^{T} w_{\mathcal{F}_t}(A_t)$ in terms of the eluder dimension of the class of functions $\mathcal{F}$. When combined with Proposition 1, this analysis provides regret bounds for both Thompson sampling and for a UCB algorithm.

## 6  Construction of confidence sets

The abstract confidence sets we construct are centered around least squares estimates $\hat{f}_t^{LS} \in \arg\min_{f \in \mathcal{F}} L_{2,t}(f)$ where $L_{2,t}(f) = \sum_1^{t-1}(f(A_t) - R_t)^2$ is the cumulative squared prediction error.[1] The sets take the form $\mathcal{F}_t := \{f \in \mathcal{F} : \|f - \hat{f}_t^{LS}\|_{2,E_t} \leq \sqrt{\beta_t}\}$ where $\beta_t$ is an appropriately chosen confidence parameter, and the empirical 2-norm $\|\cdot\|_{2,E_t}$ is defined by $\|g\|_{2,E_t}^2 = \sum_1^{t-1} g^2(A_k)$. Hence $\|f - f_\theta\|_{2,E_t}^2$ measures the cumulative discrepancy between the previous predictions of $f$ and $f_\theta$.

The following lemma is the key to constructing strong confidence sets $(\mathcal{F}_t : t \in \mathbb{N})$. For an arbitrary function $f$, it bounds the squared error of $f$ from below in terms of the empirical loss of the true function $f_\theta$ and the aggregate empirical discrepancy $\|f - f_\theta\|_{2,E_t}^2$ between $f$ and $f_\theta$. It establishes that for any function $f$, with high probability, the random process $(L_{2,t}(f) : t \in \mathbb{N})$ never falls below the process $(L_{2,t}(f_\theta) + \frac{1}{2}\|f - f_\theta\|_{2,E_t}^2 : t \in \mathbb{N})$ by more than a fixed constant. A proof of the lemma is provided in the appendix. Recall that $\eta$ is a constant given in Assumption 2.

**Lemma 1.** *For any $\delta > 0$ and $f : \mathcal{A} \mapsto \mathbb{R}$,*

$$\mathbb{P}\left(L_{2,t}(f) \geq L_{2,t}(f_\theta) + \frac{1}{2}\|f - f_\theta\|_{2,E_t}^2 - 4\eta^2 \log(1/\delta) \quad \forall t \in \mathbb{N} \,\middle|\, \theta\right) \geq 1 - \delta.$$

By Lemma 1, with high probability, $f$ can enjoy lower squared error than $f_\theta$ only if its empirical deviation $\|f - f_\theta\|_{2,E_t}^2$ from $f_\theta$ is less than $8\eta^2 \log(1/\delta)$. Through a union bound, this property holds uniformly for all functions in a finite subset of $\mathcal{F}$. To extend this result to infinite classes of functions, we measure the function class at some discretization scale $\alpha$. Let $N(\mathcal{F}, \alpha, \|\cdot\|_\infty)$ denote the $\alpha$-covering number of $\mathcal{F}$ in the sup-norm $\|\cdot\|_\infty$, and let

$$\beta_t^*(\mathcal{F}, \delta, \alpha) := 8\eta^2 \log\left(N(\mathcal{F}, \alpha, \|\cdot\|_\infty)/\delta\right) + 2\alpha t \left(8C + \sqrt{8\eta^2 \ln(4t^2/\delta)}\right). \tag{4}$$

**Proposition 2.** *For all $\delta > 0$ and $\alpha > 0$, if*

$$\mathcal{F}_t = \left\{ f \in \mathcal{F} : \left\| f - \hat{f}_t^{LS} \right\|_{2,E_t} \leq \sqrt{\beta_t^*(\mathcal{F}, \delta, \alpha)} \right\}$$

*for all $t \in \mathbb{N}$, then*

$$\mathbb{P}\left( f_\theta \in \bigcap_{t=1}^\infty \mathcal{F}_t \,\middle|\, \theta \right) \geq 1 - 2\delta.$$

**Example 3.** *Suppose $\Theta \subset [0,1]^d$ and for each $a \in \mathcal{A}$, $f_\theta(a)$ is an $L$–Lipschitz function of $\theta$. Then $N(\mathcal{F}, \alpha, \|\cdot\|_\infty) \leq (1 + L/\epsilon)^d$ and hence $\log N(\mathcal{F}, \alpha, \|\cdot\|_\infty) \leq d \log(1 + L/\epsilon)$.*

# 7   Measuring the rate at which confidence sets shrink

Our remaining task is to provide a worst case bound on the sum $\sum_1^T w_{\mathcal{F}_t}(A_t)$. First consider the case of a linearly parameterized model where $f_\rho(a) := \langle \phi(a), \rho \rangle$ for each $\rho \in \Theta \subset \mathbb{R}^d$. Then, it can be shown that our confidence set takes the form $\mathcal{F}_t := \{f_\rho : \rho \in \Theta_t\}$ where $\Theta_t \subset \mathbb{R}^d$ is an ellipsoid. When an action $A_t$ is sampled, the ellipsoid shrinks in the direction $\phi(A_t)$. Here the explicit geometric structure of the confidence set implies that the width $w_{\mathcal{F}_t}$ shrinks not only at $A_t$ but also at any other action whose feature vector is not orthogonal to $\phi(A_t)$. Some linear algebra leads to a worst case bound on $\sum_1^T w_{\mathcal{F}_t}(A_t)$. For a general class of functions, the situation is much subtler, and we need to measure the way in which the width at each action can be reduced by sampling other actions.

The following result uses our new notion of dimension to bound the number of times the width of the confidence interval for a selected action $A_t$ can exceed a threshold.

**Proposition 3.** *If $(\beta_t \geq 0 | t \in \mathbb{N})$ is a nondecreasing sequence and $\mathcal{F}_t := \{f \in \mathcal{F} : \|f - \hat{f}_t^{LS}\|_{2,E_t} \leq \sqrt{\beta_t}\}$ then with probability 1*

$$\sum_{t=1}^T \mathbf{1}(w_{\mathcal{F}_t}(A_t) > \epsilon) \leq \left( \frac{4\beta_T}{\epsilon^2} + 1 \right) \dim_E(\mathcal{F}, \epsilon)$$

*for all $T \in \mathbb{N}$ and $\epsilon > 0$.*

Using Proposition 3, one can bound the sum $\sum_{t=1}^T w_{\mathcal{F}_t}(A_t)$, as established by the following lemma. To extend our analysis to infinite classes of functions, we consider the $\alpha_T^\mathcal{F}$–eluder dimension of $\mathcal{F}$, where

$$\alpha_t^\mathcal{F} = \max\left\{ \frac{1}{t^2}, \inf\{\|f_1 - f_2\|_\infty : f_1, f_2 \in \mathcal{F}, f_1 \neq f_2\} \right\}. \tag{5}$$

**Lemma 2.** *If $(\beta_t \geq 0 | t \in \mathbb{N})$ is a nondecreasing sequence and $\mathcal{F}_t := \{f \in \mathcal{F} : \|f - \hat{f}_t^{LS}\|_{2,E_t} \leq \sqrt{\beta_t}\}$ then with probability 1, for all $T \in \mathbb{N}$,*

$$\sum_{t=1}^T w_{\mathcal{F}_t}(A_t) \leq \frac{1}{T} + \min\left\{\dim_E\left(\mathcal{F}, \alpha_T^\mathcal{F}\right), T\right\} C + 4\sqrt{\dim_E\left(\mathcal{F}, \alpha_T^\mathcal{F}\right) \beta_T T}. \tag{6}$$

# 8   Main Result

Our analysis provides a new guarantee both for Thompson sampling, and for a UCB algorithm $\pi^{\mathcal{F}_{1:\infty}^*}$ executed with appropriate confidence sets $\{\mathcal{F}_t^* : t \in \mathbb{N}\}$. Recall, for a sequence of confidence sets $\{\mathcal{F}_t : t \in \mathbb{N}\}$ we denote by $\pi^{\mathcal{F}_{1:\infty}}$ the UCB algorithm that chooses an action $\bar{A}_t \in \arg\max_{a \in \mathcal{A}} \left\{ \sup_{f \in \mathcal{F}_t} f_\theta(a) \right\}$ at each time $t$. We establish bounds that are, up to a logarithmic factor, of order

$$\sqrt{\underbrace{\dim_E\left(\mathcal{F}, T^{-2}\right)}_{\text{Eluder dimension}} \underbrace{\log\left(N\left(\mathcal{F}, T^{-2}, \|\cdot\|_\infty\right)\right)}_{\text{log–covering number}} T}.$$

This term depends on two measures of the complexity of the function class $\mathcal{F}$. The first, which controls for statistical over–fitting, grows logarithmically in the cover numbers of the function class. This is a common feature of notions of dimension from statistical learning theory. The second measure of complexity, the eluder dimension, measures the extent to which the reward value at one action can be inferred by sampling other actions.

The next two propositions, which provide finite time bounds for a particular UCB algorithm and for Thompson sampling, follow by combining Proposition 1, Propsition 2, and Lemma 2. Define,

$$\mathcal{B}(\mathcal{F}, T, \delta) = \frac{1}{T} + \left[ \min \left\{ \dim_E \left( \mathcal{F}, \alpha_T^{\mathcal{F}} \right), T \right\} \right] C + 4 \sqrt{\dim_E \left( \mathcal{F}, \alpha_T^{\mathcal{F}} \right) \beta_T^* \left( \mathcal{F}, \alpha_T^{\mathcal{F}}, \delta \right) T}.$$

Notice that $\mathcal{B}(\mathcal{F}, T, \delta)$ is the right hand side of the bound (6) with $\beta_T$ taken to be $\beta_T^*(\mathcal{F}, \alpha_T^{\mathcal{F}}, \delta)$.

**Proposition 4.** *Fix any* $\delta > 0$ *and* $T \in \mathbb{N}$, *and define for each* $t \in \mathbb{N}$, $\mathcal{F}_t^* = \left\{ f \in \mathcal{F} : \left\| f - \hat{f}_t^{LS} \right\|_{2, E_t} \leq \sqrt{\beta_t^* \left( \mathcal{F}, \alpha_T, \delta \right)} \right\}$. *Then,*

$$\mathbb{P} \left\{ \mathcal{R}(T, \pi^{\mathcal{F}_{1:\infty}^*}) \leq \mathcal{B}(\mathcal{F}, T, \delta) \mid \theta \right\} \geq 1 - 2\delta$$

$$\mathbb{E} \left[ \mathcal{R}(T, \pi^{\mathcal{F}_{1:\infty}^*}) \mid \theta \right] \leq \mathcal{B}(\mathcal{F}, T, \delta) + 2\delta TC$$

**Proposition 5.** *For any* $T \in \mathbb{N}$,

$$\mathbb{E} \left[ \mathcal{R}(T, \pi^{\mathrm{TS}}) \right] \leq \mathcal{B}(\mathcal{F}, T, T^{-1}) + 2C$$

The next two examples show how the regret bounds of Proposition 4 and 5 specialize to $d$-dimensional linear and generalized linear models. For each of these examples $\Theta \subset \mathbb{R}^d$ and each action is associated with a known feature vector $\phi(a)$. Throughout these two examples, we fix positive constants $\gamma$ and $s$ and assume that $\gamma \geq \sup_{a \in \mathcal{A}} \|\phi(a)\|$ and $s \geq \sup_{\rho \in \Theta} \|\rho\|$. For each of these examples, a bound on $\dim_E (\mathcal{F}, \epsilon)$ is provided in the supplementary material.

**Example 4. Linear Models:** *Consider the case of a* $d$-*dimensional linear model* $f_\rho(a) := \langle \phi(a), \rho \rangle$. *Then,* $\dim_E(\mathcal{F}, \epsilon) = O(d \log(1/\epsilon))$ *and* $\log N(\mathcal{F}, \epsilon, \|\cdot\|_\infty) = O(d \log(1/\epsilon))$. *Propositions 4 and 5 therefore yield* $O(d \log(1/\alpha_T^{\mathcal{F}})\sqrt{T})$ *regret bounds. Since* $\alpha_T^{\mathcal{F}} \geq T^{-2}$, *This is tight to within a factor of* $\log T$ [3], *and matches the best available bound for a linear UCB algorithm* [2].

**Example 5. Generalized Linear Models:** *Consider the case of a* $d$-*dimensional generalized linear model* $f_\theta(a) := g(\langle \phi(a), \theta \rangle)$ *where* $g$ *is an increasing Lipschitz continuous function. Set* $\overline{h} = \sup_{\tilde{\theta}, a} g'(\langle \phi(a), \tilde{\theta} \rangle)$, $\underline{h} = \inf_{\tilde{\theta}, a} g'(\langle \phi(a), \tilde{\theta} \rangle)$ *and* $r = \overline{h}/\underline{h}$. *Then,* $\log N(\mathcal{F}, \epsilon, \|\cdot\|_\infty) = O(d \log(\overline{h}/\epsilon))$ *and* $\dim_E(\mathcal{F}, \epsilon) = O(dr^2 \log(\overline{h}/\epsilon))$, *and Propositions 4 and 5 yield* $O(rd \log(\overline{h}/\alpha_T^{\mathcal{F}})\sqrt{T})$ *regret bounds. To our knowledge, this bound is a slight improvement over the strongest regret bound available for any algorithm in this setting. The regret bound of Filippi et al.* [7] *is of order* $rd \log^{3/2}(T)\sqrt{T}$.

## 9 Conclusion

In this paper, we have analyzed two algorithms, Thompson sampling and a UCB algorithm, in a very general framework, and developed regret bounds that depend on a new notion of dimension. In constructing these bounds, we have identified two factors that control the hardness of a particular multi-armed bandit problem. First, an agent's ability to quickly attain near-optimal performance depends on the extent to which the reward value at one action can be inferred by sampling other actions. However, in order to select an action the agent must make inferences about many possible actions, and an error in its evaluation of any one could result in large regret. Our second measure of complexity controls for the difficulty of maintaining appropriate confidence sets simultaneously at every action. While our bounds are nearly tight in some cases, further analysis is likely to yield stronger results in other cases. We hope, however, that our work provides a conceptual foundation for the study of such problems, and inspires further investigation.

#### Acknowledgments

The first author is supported by a Burt and Deedee McMurty Stanford Graduate Fellowship. This work was supported in part by Award CMMI-0968707 from the National Science Foundation.

## Footnotes

[1] The results can be extended to the case where the infimum of $L_{2,t}(f)$ is unattainable by selecting a function with squared prediction error sufficiently close to the infimum.

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
