[Supplementary Material · NIPS - Sample complexity - supplementary material.pdf]

# A  Proof of Regret Decompositions

**Proposition 1.** *Fix any sequence $\{\mathcal{F}_t : t \in \mathbb{N}\}$, where $\mathcal{F}_t \subset \mathcal{F}$ is measurable with respect to $\sigma(H_t)$. Then for any $T \in \mathbb{N}$, with probability 1,*

$$\mathcal{R}(T, \pi^{\mathcal{F}_{1:\infty}}) \leq \sum_{t=1}^{T} [w_{\mathcal{F}_t}(A_t) + C\mathbf{1}(f_\theta \notin \mathcal{F}_t)] \tag{7}$$

$$\mathbb{E}\left[\mathcal{R}(T, \pi^{\mathrm{TS}})\right] \leq \mathbb{E}\sum_{t=1}^{T} [w_{\mathcal{F}_t}(A_t) + C\mathbf{1}(f_\theta \notin \mathcal{F}_t)]. \tag{8}$$

*Proof.* To reduce notation, define the upper and lower bounds $U_t(a) = \sup\{f(a) : f \in \mathcal{F}_t\}$ and $L_t(a) = \inf\{f(a) : f \in \mathcal{F}_t\}$. Whenever $f_\theta \in \mathcal{F}_t$, the bounds $L_t(a) \leq f_\theta(a) \leq U_t(a)$ hold for all actions. This implies

$$f_\theta(A_t^*) - f_\theta(A_t) \leq U_t(A_t^*) - L_t(A_t) + C\mathbf{1}(f_\theta \notin \mathcal{F}_t) \tag{9}$$

$$= w_{\mathcal{F}_t}(A_t) + C\mathbf{1}(f_\theta \notin \mathcal{F}_t) + [U_t(A_t^*) - U_t(A_t)]. \tag{10}$$

Equation (7) follows almost immediately, since the policy $\pi^{\mathcal{F}_{1:\infty}}$ chooses an action $A_t$ that maximizes $U_t(a)$. This implies $U_t(A_t) \geq U_t(A_t^*)$ by definition, and the last term in (10) is negative. The result (7) follows by summing over $t$.

Now consider equation (8). Summing equation (10) over $t$ shows,

$$\mathcal{R}(T, \pi^{\mathrm{TS}}) \leq \sum_{t=1}^{T} [w_{\mathcal{F}_t}(A_t) + C\mathbf{1}(f_\theta \notin \mathcal{F}_t)] + M_T \tag{11}$$

where $M_T := \sum_{t=1}^{T} [U_t(A_t^*) - U_t(A_t)]$. Now, by the definition of Thompson sampling $\mathbb{P}(A_t \in \cdot|H_t) = \mathbb{P}(A_t^* \in \cdot|H_t)$. That is $A_t$ and $A_t^*$ are identically distributed under the posterior. In addition, since the confidence set $\mathcal{F}_t$ is $\sigma(H_t)$–measurable, so is the induced upper confidence bound $U_t(\cdot)$. This implies $\mathbb{E}[U_t(A_t)|H_t] = \mathbb{E}[U_t(A_t^*)|H_t]$, and therefore that $\mathbb{E}[M_T] = 0$. $\square$

# B  Proof of Confidence bound

## B.1  Preliminaries: Martingale Exponential Inequalities

Consider random variables $(Z_n|n \in \mathbb{N})$ adapted to the filtration $(\mathcal{H}_n : n = 0, 1, ...)$. Assume $\mathbb{E}\left[\exp\{\lambda Z_i\}\right]$ is finite for all $\lambda$. Define the conditional mean $\mu_i = \mathbb{E}\left[Z_i \mid \mathcal{H}_{i-1}\right]$. We define the conditional cumulant generating function of the centered random variable $[Z_i - \mu_i]$ by $\psi_i(\lambda) = \log \mathbb{E}\left[\exp\left(\lambda\left[Z_i - \mu_i\right]\right) \mid \mathcal{H}_{i-1}\right]$. Let

$$M_n(\lambda) = \exp\left\{\sum_{i=1}^{n} \lambda\left[Z_i - \mu_i\right] - \psi_i(\lambda)\right\}.$$

**Lemma 3.** $(M_n(\lambda)|n \in \mathbb{N})$ *is a Martinagale, and* $\mathbb{E}\left[M_n(\lambda)\right] = 1$.

*Proof.* By definition

$$\mathbb{E}[M_1(\lambda)|\mathcal{H}_0] = \mathbb{E}[\exp\{\lambda\left[Z_1 - \mu_1\right] - \psi_1(\lambda)|\mathcal{H}_0\}] = \mathbb{E}[\exp\{\lambda\left[Z_1 - \mu_1\right]\}|\mathcal{H}_0]/\exp\{\psi_1(\lambda)\} = 1.$$

Then, for any $n \geq 2$,

$$\mathbb{E}\left[M_n(\lambda) \mid \mathcal{H}_{n-1}\right] = \mathbb{E}\left[\exp\left\{\sum_{i=1}^{n-1} \lambda\left[Z_i - \mu_i\right] - \psi_i(\lambda)\right\}\exp\{\lambda\left[Z_n - \mu_n\right] - \psi_n(\lambda)\} \mid \mathcal{H}_{n-1}\right]$$

$$= \exp\left\{\sum_{i=1}^{n-1} \lambda\left[Z_i - \mu_i\right] - \psi_i(\lambda)\right\} \mathbb{E}\left[\exp\{\lambda\left[Z_n - \mu_n\right] - \psi_n(\lambda)\} \mid \mathcal{H}_{n-1}\right]$$

$$= \exp\left\{\sum_{i=1}^{n-1} \lambda\left[Z_i - \mu_i\right] - \psi_i(\lambda)\right\} = M_{n-1}(\lambda).$$

$\square$

**Lemma 4.** *For all $x \geq 0$ and $\lambda \geq 0$, $\mathbb{P}\left(\sum_1^n \lambda Z_i \leq x + \sum_1^n \left[\lambda\mu_i + \psi_i(\lambda)\right] \quad \forall n \in \mathbb{N}\right) \geq 1 - e^{-x}$.*

*Proof.* For any $\lambda$, $M_n(\lambda)$ is a martingale with $\mathbb{E}M_n(\lambda) = 1$. Therefore, for any stopping time $\tau$, $\mathbb{E}[M_{\tau \wedge n}(\lambda)] = 1$. For arbitrary $x \geq 0$, define $\tau_x = \inf\{n \geq 0 \mid M_n(\lambda) \geq x\}$ and note that $\tau_x$ is a stopping time corresponding to the first time $M_n$ crosses the boundary at $x$. Then, $\mathbb{E}[M_{\tau_x \wedge n}(\lambda)] = 1$ and by Markov's inequality:

$$x\mathbb{P}\left(M_{\tau_x \wedge n}(\lambda) \geq x\right) \leq \mathbb{E}M_{\tau_x \wedge n}(\lambda) = 1.$$

We note that the event $\{M_{\tau_x \wedge n}(\lambda) \geq x\} = \bigcup_{k=1}^n \{M_k(\lambda) \geq x\}$. So we have shown that for all $x \geq 0$ and $n \geq 1$

$$\mathbb{P}\left(\bigcup_{k=1}^n \{M_k(\lambda) \geq x\}\right) \leq \frac{1}{x}.$$

Taking the limit as $n \to \infty$, and applying the monotone convergence theorem shows $\mathbb{P}\left(\bigcup_{k=1}^\infty \{M_k(\lambda) \geq x\}\right) \leq \frac{1}{x}$, Or, $\mathbb{P}\left(\bigcup_{k=1}^\infty \{M_k(\lambda) \geq e^x\}\right) \leq e^{-x}$. This then shows, using the definition of $M_k(\lambda)$, that

$$\mathbb{P}\left(\bigcup_{n=1}^\infty \left\{\sum_{i=1}^n \lambda[Z_i - \mu_i] - \psi_i(\lambda) \geq x\right\}\right) \leq e^{-x}.$$

$\square$

## B.2   Proof of Lemma 1

**Lemma 1.** *For any $\delta > 0$ and $f : \mathcal{A} \mapsto \mathbb{R}$,*

$$\mathbb{P}\left(L_{2,t}(f) \geq L_{2,t}(f_\theta) + \frac{1}{2}\|f - f_\theta\|_{2,E_t}^2 - 4\eta^2 \log(1/\delta) \quad \forall t \in \mathbb{N} \,\Big|\, \theta\right) \geq 1 - \delta.$$

We will transform our problem in order to apply the general exponential martingale result shown above. since we work conditionally on $\theta$, to reduce notation we denote the conditional probability and expectation operators $\mathbb{P}_\theta(\cdot) = \mathbb{P}(\cdot|\theta)$ and $\mathbb{E}_\theta(\cdot) = \mathbb{E}(\cdot|\theta)$. We set $\mathcal{H}_{t-1}$ to be the $\sigma$-algebra generated by $(H_t, A_t)$ and set $\mathcal{H}_0 = \sigma(\emptyset, \Omega)$. By previous assumptions, $\epsilon_t := R_t - f_\theta(A_t)$ satisfies $\mathbb{E}_\theta[\epsilon_t|\mathcal{H}_{t-1}] = 0$ and $\mathbb{E}_\theta[\exp\{\lambda\epsilon_t\} \mid \mathcal{H}_{t-1}] \leq \exp\left\{\frac{\lambda^2\eta^2}{2}\right\}$ a.s. for all $\lambda$. Define $Z_t = (f_\theta(A_t) - R_t)^2 - (f(A_i) - R_t)^2$.

*Proof.* By definition $\sum_1^T Z_t = L_{2,T+1}(f_\theta) - L_{2,T+1}(f)$. Some calculation shows that $Z_t = -(f(A_t) - f_\theta(A_t))^2 + 2(f(A_t) - f_\theta(A_t))\epsilon_t$. Therefore, the conditional mean and conditional cumulant generating function satisfy:

$$
\begin{aligned}
\mu_t &= \mathbb{E}_\theta[Z_t \mid \mathcal{H}_{t-1}] = -(f(A_t) - f_\theta(A_t))^2 \\
\psi_t(\lambda) &= \log\mathbb{E}_\theta[\exp(\lambda[Z_t - \mu_t]) \mid \mathcal{H}_{t-1}] \\
&= \log\mathbb{E}_\theta[\exp(2\lambda(f(A_t) - f_\theta(A_t))\epsilon_t) \mid \mathcal{H}_{t-1}] \leq \frac{(2\lambda[f(A_t) - f_\theta(A_t)])^2\eta^2}{2}
\end{aligned}
$$

Applying Lemma 4 shows that for all $x \geq 0$, $\lambda \geq 0$

$$\mathbb{P}_\theta\left(\sum_{k=1}^t \lambda Z_k \leq x - \lambda\sum_{k=1}^t (f(A_k) - f_\theta(A_k))^2 + \frac{\lambda^2}{2}(2f(A_k) - 2f_\theta(A_k))^2\eta^2 \quad \forall t \in \mathbb{N}\right) \geq 1 - e^{-x}.$$

Or, rearranging terms

$$\mathbb{P}_\theta\left(\sum_{k=1}^t Z_k \leq \frac{x}{\lambda} + \sum_{k=1}^t (f(A_k) - f_\theta(A_k))^2(2\lambda\eta^2 - 1) \quad \forall t \in \mathbb{N}\right) \geq 1 - e^{-x}.$$

Choosing $\lambda = \frac{1}{4\eta^2}$, $x = \log\frac{1}{\delta}$, and using the definition of $\sum_1^t Z_k$ implies

$$\mathbb{P}_\theta\left(L_{2,t}(f) \geq L_{2,t}(f_\theta) + \frac{1}{2}\|f - f_\theta\|_{2,E_t}^2 - 4\eta^2 \log(1/\delta) \quad \forall t \in \mathbb{N}\right) \geq 1 - \delta.$$

$\square$

## B.3 Least Squares Bound - Proof of Proposition 2

**Proposition 2.** *For all $\delta > 0$ and $\alpha > 0$, if $\mathcal{F}_t = \left\{ f \in \mathcal{F} : \left\| f - \hat{f}_t^{LS} \right\|_{2,E_t} \leq \sqrt{\beta_t^*\left(\mathcal{F}, \delta, \alpha\right)} \right\}$ for all $t \in \mathbb{N}$, then*

$$\mathbb{P}_\theta \left( f_\theta \in \bigcap_{t=1}^\infty \mathcal{F}_t \right) \geq 1 - 2\delta.$$

*Proof.* Let $\mathcal{F}^\alpha \subset \mathcal{F}$ be an $\alpha$–cover of $\mathcal{F}$ in the sup-norm in the sense that for any $f \in \mathcal{F}$ there exists $f^\alpha \in \mathcal{F}^\alpha$ such that $\|f^\alpha - f\|_\infty \leq \epsilon$. By a union bound, conditional on $\theta$, with probability at least $1 - \delta$,

$$L_{2,t}(f^\alpha) - L_{2,t}(f_\theta) \geq \frac{1}{2} \|f^\alpha - f_\theta\|_{2,E_t} - 4\eta^2 \log\left(|\mathcal{F}^\alpha|/\delta\right) \quad \forall t \in \mathbb{N}, \, f \in \mathcal{F}^\alpha.$$

Therefore, with probability at least $1 - \delta$, for all $t \in \mathbb{N}$ and $f \in \mathcal{F}$:

$$\begin{aligned}
L_{2,t}(f) - L_{2,t}(f_\theta) &\geq \frac{1}{2} \|f - f_\theta\|_{2,E_t}^2 - 4\eta^2 \log\left(|\mathcal{F}^\alpha|/\delta\right) \\
&+ \underbrace{\min_{f^\alpha \in \mathcal{F}^\alpha} \left\{ \frac{1}{2} \|f^\alpha - f_\theta\|_{2,E_t}^2 - \frac{1}{2} \|f - f_\theta\|_{2,E_t}^2 + L_{2,t}(f) - L_{2,t}(f^\alpha) \right\}}_{\text{Discretization Error}}.
\end{aligned}$$

Lemma 5, which we establish in the next section, asserts that with probability at least $1 - \delta$ the discretization error is bounded for all $t$ by $\alpha D_t$ where $D_t := t \left[ 8C + \sqrt{8\eta^2 \ln(4t^2/\delta)} \right]$. Since the least squares estimate $\hat{f}_t^{LS}$ has lower squared error than $f_\theta$ by definition, we find with probability at least $1 - 2\delta$

$$\frac{1}{2} \left\| \hat{f}_t^{\text{LS}} - f_\theta \right\|_{2,E_t}^2 \leq 4\eta^2 \log\left(|\mathcal{F}^\alpha|/\delta\right) + \alpha D_t.$$

Taking the infimum over the size of $\alpha$ covers implies:

$$\left\| \hat{f}_t^{LS} - f_\theta \right\|_{2,E_t} \leq \sqrt{8\eta^2 \log\left(N(\mathcal{F}, \alpha, \|\cdot\|_\infty)/\delta\right) + 2\alpha D_t} \overset{\text{def}}{=} \sqrt{\beta_t^*(\mathcal{F}, \delta, \alpha)}.$$

$\square$

## B.4 Discretization Error

**Lemma 5.** *If $f^\alpha$ satisfies $\|f - f^\alpha\|_\infty \leq \alpha$, then, conditional on $\theta$, with probability at least $1 - \delta$,*

$$\left| \frac{1}{2} \|f^\alpha - f_\theta\|_{2,E_t}^2 - \frac{1}{2} \|f - f_\theta\|_{2,E_t}^2 + L_{2,t}(f) - L_{2,t}(f^\alpha) \right| \leq \alpha t \left[ 8C + \sqrt{8\eta^2 \ln(4t^2/\delta)} \right] \quad \forall t \in \mathbb{N} \quad (12)$$

*Proof.* Since any two functions in $f, f^\alpha \in \mathcal{F}$ satisfy $\|f - f^\alpha\|_\infty \leq C$, it is enough to consider $\alpha \leq C$. We find

$$\left| (f^\alpha)^2(a) - (f)^2(a) \right| \leq \max_{-\alpha \leq y \leq \alpha} \left| (f(a) + y)^2 - f(a)^2 \right| = 2f(a)\alpha + \alpha^2 \leq 2C\alpha + \alpha^2$$

which implies

$$\begin{aligned}
\left| (f^\alpha(a) - f_\theta(a))^2 - (f(a) - f_\theta(a))^2 \right| &= \left| \left[ (f^\alpha)(a)^2 - f(a)^2 \right] + 2f_\theta(a)(f(a) - f^\alpha(a)) \right| \leq 4C\alpha + \alpha^2 \\
\left| (R_t - f(a))^2 - (R_t - f^\alpha(a))^2 \right| &= \left| 2R_t(f^\alpha(a) - f(a)) + f(a)^2 - f^\alpha(a)^2 \right| \leq 2\alpha|R_t| + 2C\alpha + \alpha^2
\end{aligned}$$

Summing over $t$, we find that the left hand side of (12) is bounded by

$$\sum_{k=1}^{t-1} \left( \frac{1}{2} \left[ 4C\alpha + \alpha^2 \right] + \left[ 2\alpha|R_k| + 2C\alpha + \alpha^2 \right] \right) \leq \alpha \sum_{k=1}^{t-1} \left( 6C + 2|R_k| \right)$$

Because $\epsilon_k$ is sub-Gaussian, $\mathbb{P}_\theta \left( |\epsilon_k| > \sqrt{2\eta^2 \ln(2/\delta)} \right) \leq \delta$. By a union bound, $\mathbb{P}_\theta \left( \exists k \, s.t. \, |\epsilon_k| > \sqrt{2\eta^2 \ln(4k^2/\delta)} \right) \leq \frac{\delta}{2} \sum_1^\infty \frac{1}{k^2} \leq \delta$. Since $|R_k| \leq C + |\epsilon_k|$ this shows that with probability at least $1 - \delta$ the discretization error is bounded for all $t$ by $\alpha D_t$ where $D_t := t \left[ 8C + 2\sqrt{2\eta^2 \ln(4t^2/\delta)} \right]$. $\square$

## C  Bounding the sum of widths

**Proposition 3.** *If $(\beta_t \geq 0 | t \in \mathbb{N})$ is a nondecreasing sequence and $\mathcal{F}_t := \{f \in \mathcal{F} : \|f - \hat{f}_t^{LS}\|_{2, E_t} \leq \sqrt{\beta_t}\}$ then*

$$\sum_{t=1}^{T} \mathbf{1}(w_{\mathcal{F}_t}(A_t) > \epsilon) \leq \left(\frac{4\beta_T}{\epsilon^2} + 1\right) \dim_E(\mathcal{F}, \epsilon)$$

*for all $T \in \mathbb{N}$ and $\epsilon > 0$.*

*Proof.* We begin by showing that, for $t \leq T$, if $w_t(A_t) > \epsilon$ then $A_t$ is $\epsilon$-dependent on fewer than $4\beta_T / \epsilon^2$ disjoint subsequences of $(A_1, .., A_{t-1})$. To see this, note that if $w_{\mathcal{F}_t}(A_t) > \epsilon$ there are $\overline{f}, \underline{f} \in \mathcal{F}_t$ such that $\overline{f}(A_t) - \underline{f}(A_t) > \epsilon$. By definition, since $\overline{f}(A_t) - \underline{f}(A_t) > \epsilon$, if $A_t$ is $\epsilon$-dependent on a subsequence $(A_{i_1}, .., A_{i_k})$ of $(A_1, .., A_{t-1})$ then $\sum_{j=1}^{k} (\overline{f}(A_{i_j}) - \underline{f}(A_{i_j}))^2 > \epsilon^2$. It follows that, if $A_t$ is $\epsilon$-dependent on $K$ disjoint subsequences of $(A_1, .., A_{t-1})$ then $\|\overline{f} - \underline{f}\|_{2, E_t}^2 > K\epsilon^2$. By the triangle inequality, we have

$$\left\|\overline{f} - \underline{f}\right\|_{2, E_t} \leq \left\|\overline{f} - \hat{f}_t^{LS}\right\|_{2, E_t} + \left\|\underline{f} - \hat{f}_t^{LS}\right\|_{2, E_t} \leq 2\sqrt{\beta_t} \leq 2\sqrt{\beta_T}.$$

and it follows that $K < 4\beta_T / \epsilon^2$.

Next, we show that in any action sequence $(a_1, .., a_\tau)$, there is some element $a_j$ that is $\epsilon$-dependent on at least $\tau/d - 1$ disjoint subsequences of $(a_1, .., a_{j-1})$, where $d := \dim_E(\mathcal{F}, \epsilon)$. To show this, for an integer $K$ satisfying $Kd + 1 \leq \tau \leq Kd + d$, we will construct $K$ disjoint subsequences $B_1, \ldots, B_K$. First let $B_i = (a_i)$ for $i = 1, .., K$. If $a_{K+1}$ is $\epsilon$-dependent on each subsequence $B_1, .., B_K$, our claim is established. Otherwise, select a subsequence $B_i$ such that $a_{K+1}$ is $\epsilon$-independent and append $a_{K+1}$ to $B_i$. Repeat this process for elements with indices $j > K + 1$ until $a_j$ is $\epsilon$-dependent on each subsequence or $j = \tau$. In the latter scenario $\sum |B_i| \geq Kd$, and since each element of a subsequence $B_i$ is $\epsilon$-independent of its predecessors, $|B_i| = d$. In this case, $a_\tau$ must be $\epsilon$-dependent on each subsequence, by the definition of $\dim_E(\mathcal{F}, \epsilon)$.

Now consider taking $(a_1, .., a_\tau)$ to be the subsequence $(A_{t_1}, \ldots, A_{t_\tau})$ of $(A_1, \ldots, A_T)$ consisting of elements $A_t$ for which $w_{\mathcal{F}_t}(A_t) > \epsilon$. As we have established, each $A_{t_j}$ is $\epsilon$-dependent on fewer than $4\beta_T / \epsilon^2$ disjoint subsequences of $(A_1, .., A_{t_j - 1})$. It follows that each $a_j$ is $\epsilon$-dependent on fewer than $4\beta_T / \epsilon^2$ disjoint subsequences of $(a_1, .., a_{j-1})$. Combining this with the fact we have established that there is some $a_j$ that is $\epsilon$-dependent on at least $\tau/d - 1$ disjoint subsequences of $(a_1, .., a_{j-1})$, we have $\tau/d - 1 \leq 4\beta_T / \epsilon^2$. It follows that $\tau \leq (4\beta_T / \epsilon^2 + 1)d$, which is our desired result. $\qquad\square$

**Lemma 2.** *If $(\beta_t \geq 0 | t \in \mathbb{N})$ is a nondecreasing sequence and $\mathcal{F}_t := \{f \in \mathcal{F} : \|f - \hat{f}_t^{LS}\|_{2, E_t} \leq \sqrt{\beta_t}\}$ then with probability 1,*

$$\sum_{t=1}^{T} w_{\mathcal{F}_t}(A_t) \leq \frac{1}{T} + \min\left\{\dim_E\left(\mathcal{F}, \alpha_T^{\mathcal{F}}\right), T\right\} C + 4\sqrt{\dim_E\left(\mathcal{F}, \alpha_T^{\mathcal{F}}\right)\beta_T T} \tag{13}$$

*for all $T \in \mathbb{N}$.*

*Proof.* To reduce notation, write $d = \dim_E\left(\mathcal{F}, \alpha_T^{\mathcal{F}}\right)$ and $w_t = w_t(A_t)$. Reorder the sequence $(w_1, ..., w_T) \to (w_{i_1}, ..., w_{i_T})$ where $w_{i_1} \geq w_{i_2} \geq ... \geq w_{i_T}$. We have

$$\sum_{t=1}^{T} w_{\mathcal{F}_t}(A_t) = \sum_{t=1}^{T} w_{i_t} = \sum_{t=1}^{T} w_{i_t} \mathbf{1}\left\{w_{i_t} \leq \alpha_T^{\mathcal{F}}\right\} + \sum_{t=1}^{T} w_{i_t} \mathbf{1}\left\{w_{i_t} > \alpha_T^{\mathcal{F}}\right\} \leq \frac{1}{T} + \sum_{t=1}^{T} w_{i_t} \mathbf{1}\left\{w_{i_t} > \alpha_T^{\mathcal{F}}\right\}.$$

The final step in the above inequality uses that either $\alpha_T^{\mathcal{F}} = T^{-2}$ and $\sum_{t=1}^{T} \alpha_T^{\mathcal{F}} = T^{-1}$ or $\alpha_T^{\mathcal{F}}$ is set below the smallest possible width and hence $\mathbf{1}\left\{w_{i_t} \leq \alpha_T^{\mathcal{F}}\right\}$ never occurs.

Now, we know $w_{i_t} \leq C$. In addition, $w_{i_t} > \epsilon \iff \sum_{k=1}^{T} \mathbf{1}\left(w_{\mathcal{F}_k}(A_k) > \epsilon\right) \geq t$. By Proposition 3, this can only occur if $t < \left(\frac{4\beta_T}{\epsilon^2} + 1\right) \dim_E(\mathcal{F}, \epsilon)$. For $\epsilon \geq \alpha_T^{\mathcal{F}}$, $\dim_E(\mathcal{F}, \epsilon) \leq \dim_E(\mathcal{F}, \alpha_T^{\mathcal{F}}) = d$, since $\dim_E(\mathcal{F}, \epsilon')$ is nonincreasing in $\epsilon'$. Therefore, when $w_{i_t} > \epsilon \geq \alpha_T^{\mathcal{F}}$, $t \leq \left(\frac{4\beta_T}{\epsilon^2} + 1\right)d$ which implies $\epsilon \leq \sqrt{\frac{4\beta_T d}{t - d}}$. This shows that if $w_{i_t} > \alpha_T^{\mathcal{F}}$, then $w_{i_t} \leq \min\left\{C, \sqrt{\frac{4\beta_T d}{t - d}}\right\}$. Therefore,

$$\sum_{t=1}^{T} w_{i_t} \mathbf{1}\left\{w_{i_t} > \alpha_T^{\mathcal{F}}\right\} \leq dC + \sum_{t=d+1}^{T} \sqrt{\frac{4d\beta_T}{t - d}} \leq dC + 2\sqrt{d\beta_T} \int_{t=0}^{T} \frac{1}{\sqrt{t}} dt = dC + 4\sqrt{d\beta_T T}.$$

To complete the proof, we combine this with the fact that the sum of widths is always bounded by $CT$. This implies:

$$\sum_{t=1}^{T} w_{\mathcal{F}_t}(A_t) \leq \min\left\{TC, \; \frac{1}{T} + \dim_E\left(\mathcal{F}, \alpha_T^{\mathcal{F}}\right)C, +4\sqrt{\dim_E\left(\mathcal{F}, \alpha_T^{\mathcal{F}}\right)\beta_T T}\right\}$$

$$\leq \frac{1}{T} + \min\left\{\dim_E\left(\mathcal{F}, \alpha_T^{\mathcal{F}}\right)C, \; TC\right\} + 4\sqrt{\dim_E\left(\mathcal{F}, \alpha_T^{\mathcal{F}}\right)\beta_T T}$$

$\square$

# D   Bounds on Eluder Dimension for Common Function Classes

Definition 4, which defines the eluder dimension of a class of functions, can be equivalently written as follows. The $\epsilon$-eluder dimension of a class of functions $\mathcal{F}$ is the length of the longest sequence $a_1, .., a_\tau$ such that for some $\epsilon' \geq \epsilon$

$$w_k := \sup\left\{(f_{\rho_1} - f_{\rho_2})(a_k) : \sqrt{\sum_{i=1}^{k-1}(f_{\rho_1} - f_{\rho_2})^2(a_i)} \leq \epsilon' \; \rho_1, \rho_2 \in \Theta\right\} > \epsilon' \qquad (14)$$

for each $k \leq \tau$.

## D.1   Finite Action Spaces

Any action is $\epsilon'$–dependent on itself since $\sup\left\{(f_{\rho_1} - f_{\rho_1})(a) : \sqrt{(f_{\rho_1} - f_{\rho_2})^2(a)} \leq \epsilon' \; \rho_1, \rho_2 \in \Theta\right\} \leq \epsilon'$. Therefore, for all $\epsilon > 0$, the $\epsilon$-eluder dimension of $\mathcal{A}$ is bounded by $|\mathcal{A}|$.

## D.2   Linear Case

**Proposition 6.** *Suppose $\Theta \subset \mathbb{R}^d$ and $f_\theta(a) = \theta^T\phi(a)$. Assume there exist constants $\gamma$, and $S$, such that for all $a \in \mathcal{A}$ and $\rho \in \Theta$, $\|\rho\|_2 \leq S$, and $\|\phi(a)\|_2 \leq \gamma$. Then $\dim_E(\mathcal{F}, \epsilon) \leq 3d\frac{e}{e-1}\ln\left\{3 + 3\left(\frac{2S}{\epsilon}\right)^2\right\} + 1$.*

To simplify the notation, define $w_k$ as in (14), $\phi_k = \phi(a_k)$, $\rho = \rho_1 - \rho_2$, and $\Phi_k = \sum_{i=1}^{k-1}\phi_i\phi_i^T$. In this case, $\sum_{i=1}^{k-1}(f_{\rho_1} - f_{\rho_2})^2(a_i) = \rho^T\Phi_k\rho$, and by the triangle inequality $\|\rho\|_2 \leq 2S$. The proof follows by bounding the number of times $w_k > \epsilon'$ can occur.

**Step 1:** If $w_k \geq \epsilon'$ then $\phi_k^T V_k^{-1}\phi_k \geq \frac{1}{2}$ where $V_k := \Phi_k + \lambda I$ and $\lambda = \left(\frac{\epsilon'}{2S}\right)^2$.

*Proof.* We find $w_k \leq \max\left\{\rho^T\phi_k : \rho^T\Phi_k\rho \leq (\epsilon')^2, \; \rho^T I\rho \leq (2S)^2\right\} \leq \max\left\{\rho^T\phi_k : \rho^T V_k\rho \leq 2(\epsilon')^2\right\} = \sqrt{2(\epsilon')^2}\|\phi_k\|_{V_k^{-1}}$. The second inequality follows because any $\rho$ that is feasible for the first maximization problem must satisfy $\rho^T V_k\rho \leq (\epsilon')^2 + \lambda(2S)^2 = 2(\epsilon')^2$. By this result, $w_k \geq \epsilon'$ implies $\|\phi_k\|_{V_k^{-1}}^2 \geq 1/2$. $\square$

**Step 2:** If $w_i \geq \epsilon'$ for each $i < k$ then $\det V_k \geq \lambda^d\left(\frac{3}{2}\right)^{k-1}$ and $\det V_k \leq \left(\frac{\gamma^2(k-1)}{d} + \lambda\right)^d$.

*Proof.* Since $V_k = V_{k-1} + \phi_k\phi_k^T$, using the Matrix Determinant Lemma,

$$\det V_k = \det V_{k-1}\left(1 + \phi_k^T V_k^{-1}\phi_k\right) \geq \det V_{k-1}\left(\frac{3}{2}\right) \geq ... \geq \det[\lambda I]\left(\frac{3}{2}\right)^{k-1} = \lambda^d\left(\frac{3}{2}\right)^{k-1}.$$

Recall that $\det V_k$ is the product of the eigenvalues of $V_k$, whereas $\text{trace}[V_k]$ is the sum. As noted in [1], $\det V_k$ is maximized when all eigenvalues are equal. This implies: $\det V_k \leq \left(\frac{\text{trace}[V_k]}{d}\right)^d \leq \left(\frac{\gamma^2(t-1)}{d} + \lambda\right)^d$. $\square$

**Step 3:** Complete Proof

*Proof.* Manipulating the result of Step 2 shows $k$ must satisfy the inequality: $\left(\frac{3}{2}\right)^{\frac{k-1}{d}} \leq \alpha_0 \left[\frac{k-1}{d}\right] + 1$ where $\alpha_0 = \left(\frac{\gamma^2}{\lambda}\right) = \left(\frac{2S\gamma}{\epsilon'}\right)^2$. Let $B(x, \alpha) = \max\left\{B : (1+x)^B \leq \alpha B + 1\right\}$. The number of times $w_k > \epsilon'$ can occur is bounded by $dB(1/2, \alpha_0) + 1$.

We now derive an explicit bound on $B(x, \alpha)$ for any $x \leq 1$. Note that any $B \geq 1$ must satisfy the inequality: $\ln\{1+x\}B \leq \ln\{1+\alpha\} + \ln B$. Since $\ln\{1+x\} \geq x/(1+x)$, using the transformation of variables $y = B\left[x/(1+x)\right]$ gives:

$$y \leq \ln\{1+\alpha\} + \ln\frac{1+x}{x} + \ln y \leq \ln\{1+\alpha\} + \ln\frac{1+x}{x} + \frac{y}{e} \implies y \leq \frac{e}{e-1}\left(\ln\{1+\alpha\} + \ln\frac{1+x}{x}\right).$$

This implies $B(x, \alpha) \leq \frac{1+x}{x}\frac{e}{e-1}\left(\ln\{1+\alpha\} + \ln\frac{1+x}{x}\right)$. The claim follows by plugging in $\alpha = \alpha_0$ and $x = 1/2$.

$\square$

## D.3   Generalized Linear Models

**Proposition 7.** *Suppose $\Theta \subset \mathbb{R}^d$ and $f_\theta(a) = g(\theta^T\phi(a))$ where $g(\cdot)$ is a differentiable and strictly increasing function. Assume there exist constants $\underline{h}$, $\overline{h}$, $\gamma$, and $S$, such that for all $a \in \mathcal{A}$ and $\rho \in \Theta$, $0 < \underline{h} \leq g'(\rho^T\phi(a)) \leq \overline{h}$, $\|\rho\|_2 \leq S$, and $\|\phi(a)\|_2 \leq \gamma$. Then $\dim_E(\mathcal{F}, \epsilon) \leq 3dr^2\frac{e}{e-1}\ln\left\{3r^2 + 3r^2\left(\frac{2S\overline{h}}{\epsilon}\right)^2\right\} + 1$.*

The proof follows three steps which closely mirror those used to prove Proposition 6.

**Step 1:** If $w_k \geq \epsilon'$ then $\phi_k^T V_k^{-1}\phi_k \geq \frac{1}{2r^2}$ where $V_k := \Phi_k + \lambda I$ and $\lambda = \left(\frac{\epsilon'}{2S\underline{h}}\right)^2$.

*Proof.* By definition $w_k \leq \max\left\{g\left(\rho^T\phi_k\right) : \sum_{i=1}^{k-1} g\left(\rho^T\phi(a_i)\right)^2 \leq (\epsilon')^2, \rho^T I \rho \leq (2S)^2\right\}$. By the uniform bound on $g'(\cdot)$ this is less than $\max\left\{\overline{h}\rho^T\phi_k : \underline{h}^2\rho^T\Phi_k\rho \leq (\epsilon')^2, \rho^T I \rho \leq (2S)^2\right\} \leq \max\left\{\overline{h}\rho^T\phi_k : \underline{h}^2\rho^T V_k\rho \leq 2(\epsilon')^2\right\} = \sqrt{2(\epsilon')^2/r^2}\|\phi_k\|_{V_k^{-1}}$.
$\square$

**Step 2:** If $w_i \geq \epsilon'$ for each $i < k$ then $\det V_k \geq \lambda^d\left(\frac{3}{2}\right)^{k-1}$ and $\det V_k \leq \left(\frac{\gamma^2(k-1)}{d} + \lambda\right)^d$.

**Step 3:** Complete Proof

*Proof.* The above inequalities imply $k$ must satisfy: $\left(1 + \frac{1}{2r^2}\right)^{\frac{k-1}{d}} \leq \alpha_0\left[\frac{k-1}{d}\right]$ where $\alpha_0 = \gamma^2/\lambda$. Therefore, as in the linear case, the number of times $w_k > \epsilon'$ can occur is bounded by $dB(\frac{1}{2r^2}, \alpha_0) + 1$. Plugging these constants into the earlier bound $B(x, \alpha) \leq \frac{1+x}{x}\frac{e}{e-1}\left(\ln\{1+\alpha\} + \ln\frac{1+x}{x}\right)$ and using $1 + x \leq 3/2$ yields the result.
$\square$