[Reviews · NeurIPS 2013]

Submitted by Assigned_Reviewer_3

Summary:
*******

The paper considers the following general sequential learning problem: a set of actions \cal{A} and a parametric set F of bounded functions mapping actions to the reals are given.
At each time step, a subset \cal{A}_t \subset \cal{A} is provided, one has to choose one action A_t in \cal{A}_t, and then gets a reward R_t.
The reward is assumed to come from a sub-Gaussian distribution with mean f_\theta(A_t), where f_\theta \in F is a fixed, unknown, function.

The authors then introduce in Section 4 the notion of eluder-dimension that is designed to capture how difficult it is to infer the value of one action
from previously observed ones. A general regret decomposition is proposed in Section 5, and a specific confidence bounds around a least-squares estimate of f_\theta
are proposed in section 6.
The main result of the paper bounds the regret of a UCB-like strategy (Proposition 4) and a Thompson-like strategy (Proposition 5) in terms of the eluder-dimension of the hypothesis class F, in a distribution-free way. The eluder-dimension is then bounded in some specific examples including the case of linear and generalized linear bandits.



Details:
*******

Since the set of available actions change over time, the initially proposed setting corresponds to a sleeping bandit problem.
However it is unclear to me what are the assumptions you consider on the sets \cal{A}_t (Standard assumptions are: deterministic, stochastic, oblivious, adaptive).
It also seems to me that from section 6, they are actually assumed to be just the set \cal{A}, and especially in section 8 where the main result is clearly
stated for a "not-sleeping" bandit case ( \cal{A}_t = \cal{A} for all t).
This is actually confusing and it seems actually unclear that the main results hold as well for the general sleeping bandit setting you consider in the introduction (especially for an oblivious or adaptive choice of \cal{A}_t).
Thus I suggest for clarity that you align the proposed setting with the main result, that is either you remove the sleeping-bandit assumption entirely, or extend your results and compare them to the standard sleeping-bandit literature as well.
I have the feeling the only purpose of this general sleeping-bandit setting is to be able to handle the contextual bandit problem as a special case.


Question: what about distribution-dependent bounds scaling with O(\log(T))?


Example 2: If the noiseless binary classification setting as detailed line 202-205 indeed provides interesting motivation for introduction of the eluder-dimension,
I find on the other hand lines 205 to 212 confusing: Both the VC-dimension and the eluder-dimension are quantities that depend on a set of functions, not on a specific algorithm. Thus I am not sure to see the goal of those lines. Please clarify.

Proposition 3: It may be useful to emphasize a little more the result of this proposition, since this is really the key point that enables you to derive the main regret bounds.

Other point: In algorithm 1, you use no regularization parameter (that is you use \Phi_t and not for instance \Phi_t + \lambda I).
However in the proof (Section D.2) you introduce such lambda, for an upper bound on the max value.
It may be useful to clarify if this is only needed in the proof or if the algorithm also needs it. You may also comment on the absence of lambda w.r.t such algorithms as OFUL. Also the assumption that the bound S seems needed but is not reported.
Please clarify all these points.

Minor comments/typos:
Line 224: missing "of.
Line 228: mention that the label of a is defined to be f_\theta(a).
Line 302: I find the notation using E_t a bit awkward: what is E_t?
Lemma 1: It may be useful to recall that eta is the constant from the sub-Gaussian assumptions.
Line 518: "The implies".
Line 539: Missing “|” before \cal{H}_0
Line 682. Maybe add something like "by definition of dim_E(F,\epsilon)", or "otherwise, it would contradict the definition of d".
Line 699: 1 should be 1/T.
Line 728: "(f_{\rho_1} - f_{\rho_1})(a_k)" change second \rho_1.
Line 735: idem.
Line 706-707: To be able to deduce that, you should replace the previous arguments with w_{i_t} \geq \epsilon (instead of >) from line 698.


Quality. Good, sound.
Clarity. Good, very well written.
Originality. Good.
Significance. Good.
Summary: An interesting work, that introduces a natural notion of dimension that allows to derive regret bounds in a general MAB setting. I am hesitating between giving it a 8 or a 9.

Submitted by Assigned_Reviewer_4

This paper considers a generalized version of the multi-armed bandit with an arbitrary class F of reward functions. The authors show regret bounds for a generalized UCB and Thompson Sampling in terms of a new notion of dimension for the class F.

The paper is well written and interesting. The results are in my opinion the first reasonable step in the direction of obtaining regret bounds that 'adapts' to the difficulty of the function class F. Of course there is still a long way to go to have a complete picture: lower bounds are missing and important special cases are not covered yet. Nonetheless I am convinced that this paper will spur research in a very interesting direction and thus I recommend to accept.

Few minor comments:
- \mathcal{F}_t might not be the most appropriate notation for confidence set as it is standard to use it for filtrations (especially in the context of sequential decision making).
- Together with [4] you should cite Bubeck, Munos, Stoltz and Szepesvari 2008.
- A general reference that would fit well in 63-64 is Bubeck and Cesa-Bianchi 2012.
Summary: Interesting direction of research but still preliminary results in some sense.

Submitted by Assigned_Reviewer_5

This paper develops a regret bound that holds for both UCB algorithms and Thompson sampling, based on a new measure of the dependence among action rewards, referred to as the eluder dimension by the authors. The new regret bound matches the best available for linear models and is stronger than the best available for generalized linear models, outperforming UCB regret bounds for specific model classes.

As far as I can see, the paper is sound, clear and well-organized (Two typos in line 48 on page 1 and line 140 on page 3). It poses an important problem that may be of broad interest for NIPS audience. Their approach seems novel, the general mathematical framework looks significant.
Summary: I'm not expert enough to give a more thorough, fully-detailed review on this paper. As a general NIPS audience I find this paper interesting and solid.
Author Feedback

Author rebuttal: Thank you for the feedback. We will try to incorporate this to improve the paper.

--------------What assumptions do we make about changing action sets--------------
You're correct that we don't restrict the way these action sets change over time. This means that they could be chosen adversarially. The key assumptions are (1) the set of actions \mathcal{A}_t is revealed to the agent before an action is chosen and (2) conditional on the revealed action set, reward noise is subgaussian and has zero-mean.



--------------Changing Action Sets--------------
"It also seems to me that from section 6, they are actually assumed to be just the set \cal{A}, and especially in section 8 where the main result is clearly
stated for a "not-sleeping" bandit case ( \cal{A}_t = \cal{A} for all t)."

Unless otherwise specified, the general problem formulation from Section 2 applies to all results in the paper. This includes the results of Section 6 and those of Section 8.

A careful re-reading of section 8 reveals one possible source of confusion. There is a typo in line 372-373. Here \cal{A} should be \cal{A}_t, so that the definition here corresponds with the one given on line 138. We're sorry about this typo and any confusion it caused.

Perhaps it is somewhat surprising that the "sleeping bandit" case can be handled with so little impact on the structure of the proofs. The first key to making this work is proposition 2, which holds regardless of the actions sets change of time. Next, the confidence sets we construct are centered around least-squares estimates. Least-squares estimates depend only on the observed actions/rewards, and not on the action sets. Finally, proposition 3 holds for any possible sequence of actions in \cal{A}.



--------------Example 2--------------
The point here is to fix a hypothesis class that is "difficult" in a bandit setting (lines 205-207), but "easy" in a supervised learning setting (208-212). For this hypothesis class, it is clear that almost every action gives reward 0, which makes classification easy, but makes identifying the optimal arm hard. We should change line 206-207 to read "it is easy to see that the regret of any algorithm grows linearly with n."



--------------regularization parameter in the algorithm?--------------
The proposition established in D.2 is a bound the Eluder dimension of a particular class of functions. (Linear models with bounded parameter space) This bound is a property of a class of functions, not a particular algorithm. Lambda appears in the proof in line 751-752, where two constraints in a maximization problem are replaced with a single constraint that is a linear combination of the two. (as in weak-duality)

The uniform bound s on the norm of the parameter vector is assumed in line 404-405, right before the two examples. It is also clearly one of the conditions given in the statement of proposition 6.

If you look carefully at proposition 4 - it explicitly indicates that the result applies to the algorithm \pi^{F^*_{1:\infty}}. The confidence sets F^*_{1:\infty} are defined in the proposition, and are centered around unregularized least-squares estimates. The corresponding UCB algorithm is defined in lines 136-140 and again at the beginning of section 8.



--------------Distribution dependent bounds--------------
We don't have polished results on distribution dependent bounds - and it would be difficult to cram more technical material into such a short paper. If you're curious, we know that for a UCB algorithm, it is possible to develop distribution dependent bounds using eluder dimension. When the there is a gap between the mean reward at the best and second best arm, these regret bounds would be order log(T). Providing similar results for posterior sampling is more challenging.